# Investigation of Zhenjiang Aromatic Vinegar Production Using a Novel Dry Gelatinization Process

**DOI:** 10.3390/foods13071071

**Published:** 2024-03-31

**Authors:** Ke Wang, Yuxuan Shi, Jiaxue Feng, Yi Zhao, Hao Zhu, Di Chen, Xiaojie Gong, Meihui Fang, Yongjian Yu

**Affiliations:** Jiangsu Provincial Engineering Research Center of Grain Bioprocessing, School of Grain Science and Technology, Jiangsu University of Science and Technology, 666 Changhui Avenue, Zhenjiang 212100, China; kewang143@just.edu.cn (K.W.); 212241821210@stu.just.edu.cn (Y.S.); 212241821203@stu.just.edu.cn (J.F.); 212241821218@stu.just.edu.cn (Y.Z.); 212241821219@stu.just.edu.cn (H.Z.); 212241821201@stu.just.edu.cn (D.C.); 212241821204@stu.just.edu.cn (X.G.); 212241821303@stu.just.edu.cn (M.F.)

**Keywords:** Zhenjiang aromatic vinegar, dry gelatinization process, flavor profile, fermentation process, lactic acid

## Abstract

The traditional process of producing Zhenjiang aromatic vinegar faces challenges such as high water usage, wastewater generation, raw material losses, and limitations in mechanization and workshop conditions. This study introduces and evaluates a novel dry gelatinization process, focusing on fermentation efficiency and the vinegar flavor profile. The new process shows a 39.1% increase in alcohol conversion efficiency and a 14% higher yield than the traditional process. Vinegar produced through the dry gelatinization process has a stronger umami taste and a higher lactic acid concentration. Both processes detected 33 volatile substances, with the dry gelatinization process showing a notably higher concentration of 2-methylbutanal, which imparts a distinct fruity and chocolate aroma. These findings suggest that the dry gelatinization process outperforms the traditional process in several aspects.

## 1. Introduction

In Europe, the production of vinegar predominantly employs a liquid-state fermentation process, utilizing fruits or sugar-rich substrates. Notable examples of European vinegars encompass sherry wine vinegar, Modena balsamic vinegar, malt vinegar, and cider vinegar [1]. Conversely, in Asian nations, especially China and Japan, vinegar production is primarily characterized by the use of the traditional solid-state fermentation process [2]. Through millennia of evolution, a distinctive solid-state fermentation process has been established for traditional Chinese fermented vinegar, reflecting a significant divergence in technological approaches to vinegar production across these regions.

Zhenjiang Aromatic Vinegar (ZAV), distinguished by its unique flavor profile, holds a prominent position in the domain of Chinese vinegars. This product has achieved global recognition, with its exports reaching more than 170 countries and regions. Notably, these exports account for over 60% of the total vinegar exports from China [3]. As a paradigmatic example of traditional Chinese solid-state fermented cereal vinegars, ZAV production predominantly involves a blend of glutinous rice, wheat bran, and rice husk.

The traditional production process of ZAV includes three principal stages: alcohol fermentation, acetic acid fermentation, and post-processing [4]. In the initial alcohol fermentation stage, the primary raw material, glutinous rice, undergoes soaking in water for water absorption and starch granule expansion, followed by steaming for gelatinization. These steps aid in the ensuing liquefaction, saccharification, and alcohol fermentation processes. Post-steaming, the rice is rapidly cooled to a temperature range of 28~30 °C. Subsequently, this cooled rice is combined with wine koji, initiating the alcohol fermentation phase. Following alcohol fermentation, the resultant wine mash is mixed with wheat bran and rice husk, proceeding to a solid-state ‘layered’ acetic acid fermentation through inoculation. However, this traditional process presents several challenges: (1) significant water resource consumption during soaking and cooling; (2) substantial wastewater production, increasing treatment costs; (3) raw material losses during soaking and cooling; and (4) constraints in mechanization, high labor intensity, and less-than-ideal workshop conditions. In addressing these issues, various Chinese research institutions and vinegar production entities have explored alternative processes. Presently, numerous vinegar producers have adopted a process including raw material pulverization, cooking, liquefaction, and saccharification. Nonetheless, this modified process still requires complex machinery and extensive land and may adversely affect the vinegar flavor.

High-temperature fluidization technology, a method utilizing fluidization techniques for drying grain particles, is explored in this study [5]. This technology employs high-velocity hot air to suspend raw materials, enabling extensive contact between raw material particles and hot air, thereby facilitating rapid heat transfer to the raw materials [6]. The airflow, serving as a heating medium, supplies heat to the grain particles, causing surface moisture to evaporate and diffuse into the airflow. An uneven distribution of moisture within and outside the grain particles creates a moisture gradient, leading to rapid dehydration. This rapid loss of moisture, without sufficient time for internal moisture migration, generates compressive stress within the grain, resulting in micro-fracture formation and consequently affecting the grain’s structural integrity [7]. Key parameters that require control during the process include airflow temperature, feed rate, and drying time, with airflow temperature being the critical parameter. Recent advancements in high-temperature fluidization technology have revealed its impact on the physicochemical properties of starch [8]. Studies by Saniso et al. employing microwave-assisted hot air fluidized bed drying on rice showed that a reduction in Type-A crystal structures and the formation of Type-V structures indicate starch gelatinization [9]. Srisang et al. studied the effect of different temperatures on the quality of germinated brown rice through fluidized bed drying, finding that high-temperature fluidization created micro-fractures on the surface of the rice grains, allowing water to permeate during cooking, thus facilitating thorough starch gelatinization and reducing cooking hardness [10,11]. Li et al. investigated the moisture migration changes during the cooking of red adzuki beans, pre- and post-high-temperature fluidization treatment, using low-field nuclear magnetic resonance imaging technology. They discovered that high-temperature fluidization promotes moisture penetration and even distribution during cooking, leading to more complete starch gelatinization, thereby enhancing cooking quality [12].

Given these functionalities, the application of high-temperature fluidization technology in processing glutinous rice for aromatic vinegar brewing, achieving ‘dry gelatinization,’ could resolve the challenges faced by the traditional production process. The schematic diagram of the equipment for the dry gelatinization process is shown in Figure 1. The burner, fueled by natural gas, operates within the hot blast stove, where it combusts to generate heat. This process not only elevates the temperature of the internal air to a range of 160 to 310 °C but also facilitates a mixture with pre-heated air emanating from a dust collector (cyclone separator). Subsequently, a blower propels this heated air into a fluidization chamber. Upon traversing the fluidized bed, the air is directed through ductwork into the dust collector, wherein cyclonic separation efficiently eliminates particulate matter from the airstream. The cleansed air is then recycled back to the hot blast stove for further heating. Materials are initially stored within a hopper and await the attainment of a predetermined temperature within the fluidized bed. At this juncture, a material feeding lock mechanism is engaged, allowing the materials to be conveyed into the fluidized bed. Here, they undergo simultaneous heating and rotational movement, courtesy of the bed’s dynamics, until reaching the discharge lock mechanism for eventual expulsion. The duration of material exposure to heat—ranging from 15 to 150 s—is meticulously controlled by modulating the speed of the motor governing the rotation. The final discharge process employs both the discharge bypass airlock and a discharge port. The dry gelatinization is operated continuously, such as in this process. Since the hot air exiting the fluidized bed is ingeniously recirculated back into the hot blast stove for recombustion and subsequent reuse, this recycling mechanism significantly enhances the thermal efficiency of the equipment while concurrently minimizing energy consumption. Furthermore, the dry gelatinization process is distinguished by its brief duration of gelatinization and its elevated level of automation, presenting a compelling advantage in operational efficiency and process control [13]. Therefore, this study aims to apply high-temperature fluidization technology in the brewing process of ZAV (dry gelatinization process) and evaluate this process from the perspectives of fermentation process parameters and the flavor profile of the vinegar product.

## 2. Materials and Methods

### 2.1. Vinegar Brewing Method and Sample Preparation

This study involves the gelatinization of raw materials using both the traditional process (TP) and the dry gelatinization process (DGP). (1) Dry gelatinization treatment: A specified quantity of glutinous rice is treated in the equipment for the dry gelatinization process using 300 °C hot air for a duration of 50 s. Subsequently, the rice is cooled to room temperature for further use. (2) Traditional gelatinization treatment: An equivalent amount of glutinous rice is soaked in room-temperature water for 8–14 h, followed by pan-rinsing. The rinsed glutinous rice is then steamed in a rice steaming machine and subsequently rinsed with room-temperature water. The moisture content of the rice is measured post-rinsing. Protein, ash, fat, and starch content in glutinous rice used for fermentation were 8.73%, 0.89%, 1.12%, and 84.58% (dry base, %).

For the fermentation method and sample preparation, an appropriate amount of water is added to the dry gelatinized glutinous rice (26.8 kg glutinous rice and 13.9 kg water) to achieve a final water content equivalent to that of the traditionally gelatinized rice. The fermentation is then conducted using a traditional solid-state fermentation process. Initially, yeast powder was added to the gelatinized glutinous rice to start the alcohol fermentation, and the fermentation cycle is 7 days. Wine mash is diluted with water to achieve an alcohol concentration of approximately 9%. This base is subsequently combined with rice husk, wheat bran, and vinegar starter provided by Jiangsu Hengshun Vinegar Co., Ltd. (Zhenjiang, China) (glutinous rice:rice husk:wheat bran:vinegar starter was 1:1.7:0.8:0.5), initiating the process of stratified solid-state acetic acid fermentation. The duration of this fermentation phase is approximately 20 days. Following fermentation, the solid-state vinegar culture (termed *pei* in Chinese) is sealed for a period of 7 days. An addition of 8% roasted rice is then introduced to the *pei*, followed by leaching with water to produce raw vinegar. The raw vinegar was heated to boiling and maintained at this temperature for 30 min for sterilization. Lastly, the raw vinegar is transferred to ceramic vessels, where it undergoes a maturation process for six months, resulting in the production of the final aromatic vinegar. Notably, both the alcoholic and acetic acid fermentation processes are conducted at ambient temperature within an open environment, without any regulation of oxygen levels or humidity, to maintain traditional practices and enhance the vinegar’s complexity and depth of flavor. Marinade samples (30 mL) are collected every two days during acetic acid fermentation and sampled daily before the rotation of the fermented grains, with three replicates for each sample. The marinade sample is transferred from the bottom of the cylinder to a 50-mL centrifuge tube and stored at −20 °C. Prior to analysis, samples are thawed in a water bath at room temperature.

### 2.2. Analysis of Physicochemical Properties

5 mL of each marinade sample was used for total acidity and reducing sugar analysis. The assessment of total acidity content was conducted using titration with sodium hydroxide, adhering to the standard GB/T 12456–2008 [14]. The determination of reducing sugar content was performed employing Fehling’s reagent titration method, in accordance with GB/T 5009.7-2016 [15]. The ethanol content of wine mash after alcohol fermentation was ascertained through the distillation method.

### 2.3. Analysis of Organic Acids, Free Amino Acids, and Aroma Compounds

For the analysis of the samples from both DGP and TP, centrifugation (10 min at 6000× *g*) was conducted for marinade samples (25 mL). The resulting supernatant was treated with a mixture of 30% (*w*/*w*) ZnSO_4_ and 10.6% (*w*/*w*) K_4_[Fe(CN)_6_] to eliminate proteinaceous impurities, as described in our prior work [16]. The composition of organic acids was examined using an Agilent 1260 high-performance liquid chromatography (HPLC) system (Agilent Corp., Karlsruhe, Germany), equipped with an Aminex HPX-87H ion exclusion column (7.8 × 300 mm, i.d., 5 μm), following methodologies established in previous research [17]. Amino acid content measurement was carried out with a Sykam S-433D amino acid analyzer (Sykam GmbH, Bavaria, Germany) [18]. By evaluating the taste threshold value (TAV) of diverse amino acids in aqueous solutions, this study calculates the Dose-over-Threshold (DoT) of amino acids that impart umami, sweet, and bitter flavors in vinegar samples. The DoT is determined by the ratio of each amino acid concentration in the vinegar to its TAV in water. A DoT below 1 suggests that the amino acid does not significantly influence the flavor profile, whereas a DoT exceeding 1 indicates a substantial contribution to the overall taste sensation. The total amino acid concentrations after acetic acid fermentation were the cumulative concentrations of amino acids detected in the marinade samples. The profiling of volatile compounds was performed using headspace-solid phase microextraction (HS-SPME) coupled with gas chromatography-mass spectrometry (GC-MS), utilizing an Agilent 7890B-5977B GC-MSD system fitted with a DB-wax capillary column (30.0 m × 0.25 mm × 0.25 µm, Agilent Technology, Santa Clara, CA, USA) [19].

### 2.4. Statistical Analysis

Data were subjected to analysis of variance (ANOVA) using SPSS 20.0 and the Origin Pro 9.0 statistical software. A significance threshold was set at *p* < 0.05. All experimental procedures were replicated three times to ensure reliability.

## 3. Results and Discussion

The monitoring of key physicochemical parameters during fermentation is crucial for ensuring the success of the process. This study evaluates the feasibility of the dry gelatinization process in vinegar production, focusing on the alcohol and acetic acid fermentation stages. Comparative analyses of essential parameters, such as alcohol content post-alcohol fermentation, acidity levels during acetic acid fermentation, and reducing sugar variations, are conducted between the dry gelatinization process and the traditional process. The objective is to assess the effectiveness of the dry gelatinization process in vinegar production.

### 3.1. Alcohol Fermentation

Alcohol fermentation is a pivotal stage in vinegar production, with its efficiency significantly influencing both production efficiency and vinegar quality. As demonstrated in Table 1, the rate of alcohol fermentation in the dry gelatinization process markedly surpassed that of the traditional process, yielding a higher alcohol concentration at fermentation completion. Considering that the quality of glutinous rice and the initial moisture content of the alcohol fermentation medium for both processes were identical, the dry gelatinization process was found to surpass the traditional process by 39.1% in terms of production efficiency (alcohol production per unit time and unit of raw material). This increased efficiency was primarily due to the absence of soaking and rice sprinkling steps in the dry gelatinization process, which in the traditional process leads to starch loss. Additionally, the dry gelatinization process may facilitate enhanced starch gelatinization, creating a more conducive environment for yeast growth and reproduction. This may result in a more rapid proliferation of yeast cells, consequently reducing the duration of alcohol fermentation. These findings indicated substantial benefits of the dry gelatinization process over traditional methods in vinegar production.

### 3.2. Acetic Acid Fermentation

In the acetic acid fermentation phase, a stratified solid-state fermentation technique was utilized. Within the vinegar *pei*, microorganisms, including acetic acid bacteria and lactic acid bacteria, convert alcohol and other components from the wine mash into organic acids, notably acetic acid and lactic acid. Additionally, these microorganisms produce amino acids and volatile substances, which contribute significantly to the distinct flavor profile of the vinegar. The acetic acid fermentation process is influenced not only by external factors such as temperature and humidity but also by the composition of the wine mash [20]. Given that the dry gelatinization process modifies the gelatinization of glutinous rice, which in turn affects alcohol fermentation, it also inevitably impacts the outcome of acetic acid fermentation. Three critical parameters—total acidity, reducing sugars, and vinegar production—are identified as key indicators of acetic acid fermentation efficiency. Thus, the variations in total acidity and reducing sugar were closely monitored throughout the acetic acid fermentation phase. Figure 2 illustrates that the rate of total acidity increase in the traditional process is substantially higher compared to the dry gelatinization process during acetic acid fermentation. After 18 days, total acidity in the traditional method reached 7.178% (*w*/*v*), while in the dry gelatinization process, it attained 7.092% (*w*/*v*) after 20 days. Xu et al. (2011a) also reported that total acidity increased rapidly to 6.253% (*w*/*v*) on the 20th day after culturing in ZAV fermentation [21]. Moreover, following the sealing of the fermentation vessel, the total acidity in the dry gelatinization process is comparable to that of the traditional process. This phenomenon can be linked to the post-alcohol fermentation step, wherein water was added to the wine mash to adjust the alcohol concentration to approximately 9% (*v*/*v*). Subsequently, a mixture of wheat bran and rice husk was added to the mash in a predefined ratio according to vinegar production procedures. The dry gelatinization process, having a 14% (alcohol concentration of the dry gelatinization process—alcohol concentration of the traditional process)/alcohol concentration of the dry gelatinization process) higher alcohol concentration than the traditional process, resulted in a significantly larger initial wine mash volume post-dilution. Consequently, the initial moisture content in the vinegar mash from the dry gelatinization process was considerably higher compared to the traditional process. This disparity influenced the oxygen availability of acetic acid bacteria during acetic acid fermentation, thereby affecting acid production. This phenomenon has also been found by Fang et al., who investigated the succession patterns of bacterial communities and their correlations with environmental factors and flavor compounds during the fermentation of Zhejiang rosy vinegar [22]. After vessel sealing, anaerobic lactic acid bacteria, unaffected by oxygen levels, produced increased amounts of lactic acid. Thus, the total acid content in the dry gelatinization process surpassed that of the traditional process post-sealing. These findings indicated a need for further optimization of the dry gelatinization process to enhance acetic acid fermentation outcomes.

Figure 3 illustrates a notable increase in the concentration of reducing sugars from Day 1 to Day 11, reaching its peak on Day 11. Acetic acid fermentation, characterized by the involvement of various microorganisms, results in the enzymatic hydrolysis of starch from rice bran into reducing sugars [23]. These sugars, as key carbohydrates, contribute significantly to the distinctive flavor profile of vinegar [16]. Post-Day 11, a decline in reducing sugar levels was observed, attributable to several factors. Primarily, the fermentation process led to an increase in total acid content, causing a reduction in pH. This lowered pH environment diminished the enzymatic activity of amylase, thereby slowing down the conversion of starch to reducing sugars. Concurrently, reducing sugar was progressively consumed by ongoing fermentation processes. Furthermore, the Maillard reaction, necessitating reducing sugars as reactants, also contributed to their depletion [24]. The figure depicts similar trends in reducing sugar variation for both tested processes, within a margin of error. After completing the leaching process, the dry gelatinization process resulted in a total vinegar production of 29.4 kg, which is 14% higher than that achieved through the traditional process. Given that the quality of the raw materials utilized in both processes remains identical, and taking into account the negligible differences in the acidity levels of the vinegar produced post-leaching, this data indicates a superior yield efficiency in the dry gelatinization process when compared to the traditional process.

Although the dry gelatinization process demonstrates notable benefits over the traditional method in terms of fermentation process indicators, its effect on the flavor profile of vinegar products necessitates further evaluation. Critical to vinegar flavor are components such as free amino acids, organic acids, and volatile organic compounds. Consequently, a comparative analysis of vinegars produced by both processes was undertaken, centered on these components as pivotal parameters.

### 3.3. Composition and Variation of Free Amino Acids during Acetic Acid Fermentation

In vinegar, amino acids not only enhance the flavor profile but also offer essential nutrients. They play a crucial role in cellular metabolism regulation and function as bioactive compounds, bolstering immunity and facilitating brain development [25]. When ingested, vinegar-derived amino acids display biological and metabolic characteristics akin to free amino acids, forming complex peptides such as immunoglobulins, carrier proteins, and neurotransmitters [26]. Notably, certain amino acids in vinegar, such as histidine, methionine, cysteine, tryptophan, and tyrosine, exhibit potent antioxidant properties [27,28].

In this investigation, the evolution of free amino acids during acetic acid fermentation and their concentrations in vinegar products derived from two distinct processes were systematically monitored. The findings are presented in Figure 4 and Table 2. Across various stages of acetic acid fermentation, a total of nine free amino acids were identified in two processes. Throughout the fermentation process, there was a gradual increase in free amino acid content for both processes. Post-fermentation, total amino acid concentrations reached 87.06 mg/100 g in the dry gelatinization process and 134.53 mg/100 g in the traditional method, marking a substantial elevation from the initial levels. The augmentation of free amino acids is predominantly due to protein hydrolysis in raw materials under the action of microorganisms such as lactic acid bacteria, according to Xu et al. [21]. In the dry gelatinization process, glutamate exhibited the most significant rise, becoming the predominant amino acid post-sealing and aging, with concentrations of 27.69 and 18.60 mg/100 g, respectively. The amino acid content at the end of sealing and aging in the traditional process is only 30% of that in the dry method gelatinization process, respectively. Glutamate, known for its umami taste, implies that the dry gelatinization product may have a more intense umami flavor. In the traditional process, threonine is the amino acid whose content increases most significantly during the fermentation process, and its content is highest after the sealing and aging process. It is a sweet-tasting amino acid that is not detected in the dry gelatinization process. It is noteworthy that the overall amino acid content of vinegar post-aging is lower than after sealing. This reduction is mainly due to the Maillard reaction, a chemical interaction between reducing sugars and free amino acids during the sterilization and aging stages [29]. The variety and concentrations of free amino acids in vinegar are influenced by factors including vinegar type, raw materials, fermentation processes, and aging techniques. Comparative studies on different vinegar types reveal that glutamate is the most prevalent amino acid in both cereal and fruit vinegars [30,31,32]. Notably, the glutamate concentration in cereal vinegar (294.9 ± 17.1–887.5 ± 53.5 mg/100 g) surpasses that in fruit vinegar (373.2 ± 0.6–390.5 ± 0.02 mg/L) [33]. Research indicates that proline is most concentrated in Shanxi aged vinegar, with levels ranging from 1.59 to 3.51 mg/mL, and its total amino acid concentration escalates with aging. Chinnici et al. identified 24 amino acids in traditional balsamic vinegar, red wine vinegar, and sherry vinegar via high-performance liquid chromatography (HPLC). Among these, proline was found to be the most abundant, with concentrations of 494.67, 98.8, and 308.56 mg/kg, respectively, in the three vinegars [34].

Different amino acids contribute varying taste profiles to vinegar, with the nine amino acids detected in the dry gelatinization and traditional processes categorized into three groups: bitter (leucine, phenylalanine, valine, tyrosine), sweet (serine, threonine, ornithine), and umami (asparagine, glutamate). These amino acids collectively create a complex flavor in vinegar products. This study analyzed the taste profiles of free amino acids in vinegars produced by both processes. The DoT of these amino acids in vinegar post-aging was calculated, reflecting the ratio of amino acid concentration to their respective taste threshold values in water. The results are presented in Table 2. Upon analyzing the DoT values of umami-tasting amino acids, it was observed that the DoT values of asparagine and glutamate in the dry gelatinization vinegar surpassed those in the traditional vinegar. The total DoT contribution of umami amino acids in the dry gelatinization vinegar exceeded 74, compared to over 18 in the traditional vinegar. Thus, the dry gelatinization vinegar exhibits a more pronounced umami taste. For sweet-tasting amino acids, the DoT values of serine and threonine in the dry gelatinization vinegar were higher, whereas the DoT value of ornithine was lower than in the traditional vinegar. The total DoT contribution of sweet amino acids was over 5 in the dry gelatinization vinegar, in contrast to over 23 in the traditional vinegar. Consequently, the dry gelatinization vinegar exhibits a lower sweetness compared to the traditional vinegar. Regarding bitter-tasting amino acids, the DoT values for valine, leucine, and phenylalanine in the dry gelatinization vinegar were lower than in the traditional vinegar. The total DoT contribution of bitter amino acids was above 20 in the dry gelatinization vinegar, as opposed to over 27 in the traditional vinegar. Therefore, the dry gelatinization vinegar exhibits reduced bitterness compared to the traditional vinegar. In summary, the dry gelatinization vinegar, when evaluated from the perspective of free amino acids, demonstrates a stronger umami flavor, lesser sweetness, and reduced bitterness relative to the traditional vinegar.

### 3.4. Composition and Fluctuations of Organic Acids during Acetic Acid Fermentation

Vinegar comprises a spectrum of organic acids, ranging from volatile ones such as acetic acid to non-volatile types including lactic, malic, pyroglutamic, citric, succinic, tartaric, oxalic, and pyruvic acids. Originating from both fermentation and raw materials, acetic and lactic acids are predominant in vinegar [35], primarily generated during the acetic acid fermentation and sealing stages [21,36]. Acetic acid, typically the most abundant, imparts a robust flavor to vinegar. Conversely, lactic, tartaric, malic, and succinic acids serve as buffering agents, moderating the intensity of acetic acid and thus contributing to a more balanced flavor [31]. These organic acids are integral to the vinegar flavor profile and also function as nutritional and bioactive compounds. Certain acids, such as malic, citric, succinic, and lactic, participate in the tricarboxylic acid cycle, the final metabolic pathway for carbohydrates, lipids, and amino acids, facilitating energy production. Vinegar organic acids are widely recognized for their health benefits, including antibacterial properties, reducing fat accumulation and hyperlipidemia, improving insulin resistance and metabolic disorders, lowering high blood pressure, and mitigating fatigue [37,38]. The concentration of these organic acids in vinegar varies significantly, influenced by the vinegar type and the specifics of the fermentation process [39].

In this research, the progression of organic acids during acetic acid fermentation and their concentrations in vinegar products developed through two distinct processes were methodically monitored, with the results displayed in Figure 5. Throughout the acetic acid fermentation stages, ten organic acids (including acetic, lactic, pyruvic, oxalic, malic, tartaric, citric, succinic, a-ketoglutaric, and pyruvic acids) were identified in both processes. There was a gradual increase in the concentration of organic acids as fermentation progressed. Notably, acetic, lactic, and tartaric acids emerged as the more abundant organic acids during this phase. Acetic acid exhibited the highest relative concentration among the ten organic acids, reaching a peak of 3.85% (*w*/*v*) in the vinegar from the dry gelatinization process post-fermentation, which was marginally lower than the 4.19% (*w*/*v*) observed in the traditional process. The concentration of lactic acid in the dry gelatinization vinegar post-fermentation was 16.8% higher compared to the traditional process, whereas the tartaric acid level was 46% lower. This disparity is attributed to the water addition step in the vinegar mash post-alcohol fermentation, intended to dilute the alcohol concentration to about 9% (*v*/*v*), followed by wheat bran and rice husk mixing. According to the standard operating procedure in this laboratory, the ratio of glutinous rice to wheat bran and rice husk should be 1:1.7:0.8. Given that the alcohol content in the wine mash from the dry gelatinization process was 14% higher than in the traditional process, the volume of the former was significantly larger post-dilution. During the experimental phase, an identical ratio of wheat bran and rice husk was added to both wine mashes, leading to a notably higher initial moisture content in the dry gelatinization mash than in the traditional process. This variation impacted the oxygen availability for acetic acid bacteria during the fermentation, consequently influencing acetic acid production. However, this favored the growth and metabolism of lactic acid bacteria, resulting in an elevated concentration of lactic acid [40]. An elevated concentration of lactic acid plays a pivotal role in buffering the hydrogen ions in acetic acid, thereby mitigating the sharp taste of the latter and endowing the vinegar with a milder flavor profile [41]. In addition to acetic, lactic, and tartaric acids, the remaining six organic acids, though present in smaller quantities, are also significant contributors to the taste and aroma of vinegar. Throughout the fermentation process, these acids generally demonstrated a slow and gradual increase in concentration. Most of these compounds are intermediary metabolites in the tricarboxylic acid cycle, functioning as precursors in more intricate metabolic pathways [39]. As such, they tend not to accumulate in substantial amounts during acetic acid fermentation. The dynamics of malic acid content are particularly noteworthy, initially manifesting a decline from its peak concentration. This reduction is attributed to its metabolism by lactic acid bacteria, a process converting malic acid into lactic acid and CO_2_, referred to as malic acid-lactic acid fermentation [42]. Post-sealing and aging, it is observed that the concentrations of most organic acids diminish to varying extents. This reduction is the result of complex chemical reactions or evaporation within the system during these stages [43].

### 3.5. Composition and Fluctuations of Volatile Compounds during Acetic Acid Fermentation

The rich aroma of Zhenjiang aromatic vinegar is primarily derived from volatile substances produced by microbial metabolism and those introduced from the raw materials. Although the concentrations of these volatile compounds are generally lower than those of organic acids and amino acids, their contribution to harmonizing and enhancing the vinegar flavor profile is substantial. Research indicates that the primary volatile components in traditional Chinese vinegar encompass acids, alcohols, esters, aldehydes, ketones, and pyrazines, among others [44]. The diversity and levels of these components are affected by various factors, including the nature of raw materials, fermentation methods, yeast strains, and temperature conditions. The interactions among these volatile compounds culminate in the unique flavor characteristic of vinegar [45]. Thus, the analysis of these volatile components holds considerable importance for the quality control of vinegar production.

In this investigation, gas chromatography-mass spectrometry was employed to identify and quantify the volatile compounds in vinegar produced through two distinct processes, both pre- and post-aging (Figure 6). A total of 33 flavor compounds were detected, encompassing alcohols (6 types), acids (5 types), aldehydes and ketones (5 types), phenols (6 types), and esters (11 types). Notably, esters emerged as the most prominent volatile substances, constituting about 40% of all detected volatile compounds. These esters predominantly arise from esterification reactions between acids and alcohols [46]. Ethyl acetate, known for its fruity aroma, was observed to have the highest concentration among these compounds, a factor attributed to the significant presence of its precursor, acetic acid [44]. At the onset of acetic acid fermentation, there is a continuous production of organic acids. Concurrently, the initial stage of the vinegar mash is characterized by a significant presence of alcohol compounds, leading to the creation of various aromatic alcohols. As fermentation advances, a gradual increase in the concentration of organic acids is observed, culminating in a highly acidic environment within the vinegar mash. This acidity not only impedes the growth activity of microorganisms but also contributes to the degradation of ester compounds, thereby resulting in a reduction of ester content. Prior to aging, the concentration of ethyl acetate in vinegars from both processes was relatively similar. However, a marked decrease in ethyl acetate concentration was noted in the vinegar derived from the dry gelatinization process after aging, as compared to the traditional process vinegar.

In this study, a notable disparity was observed in the concentration of 2-methylbutanal between vinegars produced by dry gelatinization and traditional processes. The dry gelatinization process of vinegar exhibited a higher concentration of this low-abundance compound, which is characterized by its distinctive fruity and chocolate-like aroma. It is hypothesized that 2-methylbutanal is produced during the dry gelatinization of glutinous rice. This observation aligns with findings reported by Peng et al. in their study on new-style yellow wine [47]. Moreover, ethyl laurate, recognized for its fruity aroma, was absent in the dry gelatinization process vinegar, in contrast to its presence in the traditional process vinegar. Apart from these differences, other volatile substances did not show significant variations between the two types of vinegar.

## 4. Conclusions

The study introduced a dry gelatinization process to address the inefficiencies of traditional Zhenjiang aromatic vinegar production, such as significant water consumption, wastewater generation, raw material waste, and limited mechanization. This method improved alcohol production efficiency by 39.1% and vinegar yield by 14%, while also enhancing the umami flavor and increasing non-volatile organic acid levels, especially lactic acid. Despite both processes yielding 33 volatile compounds, the dry gelatinization notably boosted 2-methylbutanal levels, contributing to a distinct fruity and chocolate aroma, thus outperforming the traditional method in efficiency and flavor profile. Moreover, in the dry gelatinization process, the hot air exiting the fluidized bed is ingeniously recirculated back into the hot blast stove for recombustion and subsequent reuse, which significantly enhances the thermal efficiency of the equipment while concurrently minimizing energy consumption. Furthermore, the dry gelatinization process is distinguished by its brief duration of gelatinization and its elevated level of automation, presenting a compelling advantage in operational efficiency, process control, and economic efficiency.

## Figures and Tables

**Figure 1 foods-13-01071-f001:**
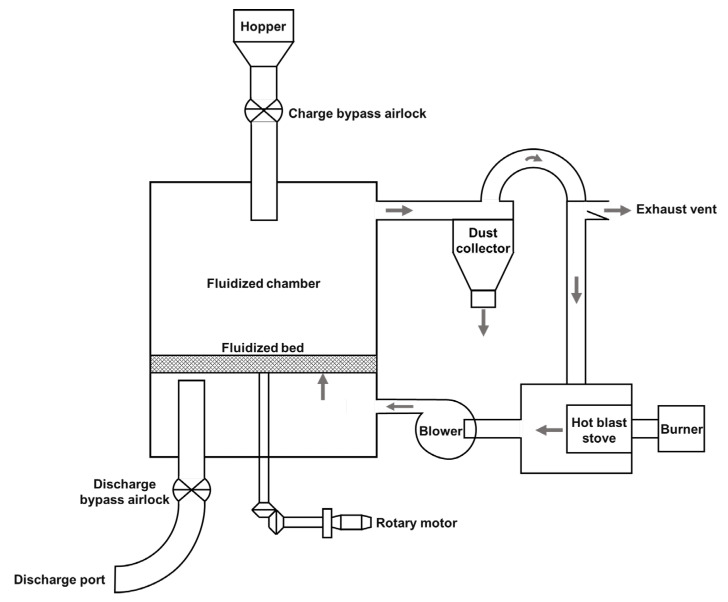
Schematic diagram of the equipment for the dry gelatinization process (The arrows indicate the direction of the hot air flow).

**Figure 2 foods-13-01071-f002:**
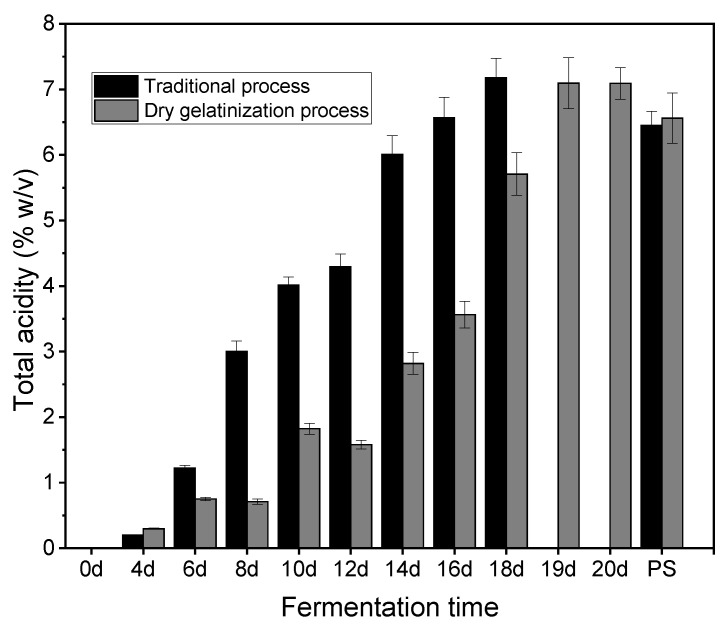
Variations in total acidity during acetic acid fermentation with the traditional process and the dry gelatinization process (PS: post-sealing).

**Figure 3 foods-13-01071-f003:**
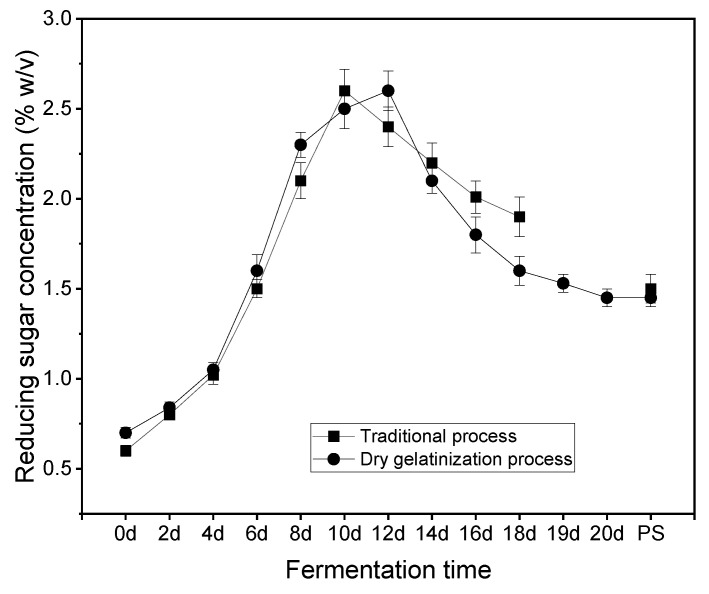
Variations in reducing sugar during acetic acid fermentation with the traditional process and the dry gelatinization process (PS: post-sealing).

**Figure 4 foods-13-01071-f004:**
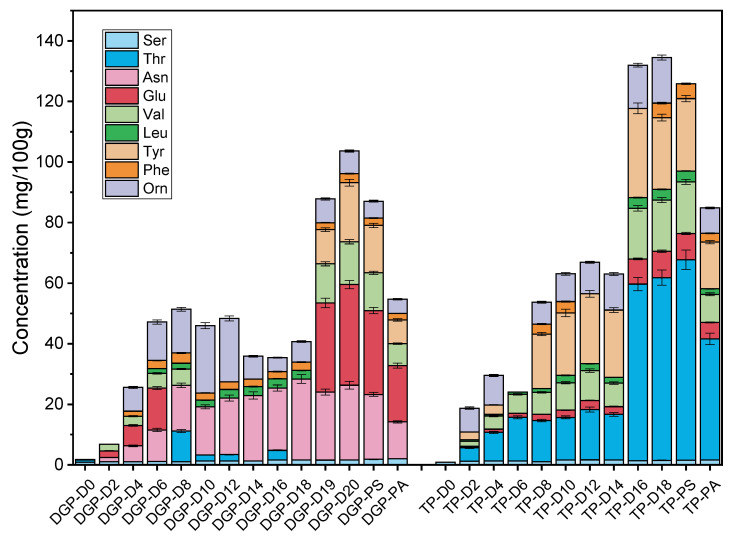
Variations in free amino acids during acetic acid fermentation with the traditional process and the dry gelatinization process (PS: post-sealing, PA: post-aging).

**Figure 5 foods-13-01071-f005:**
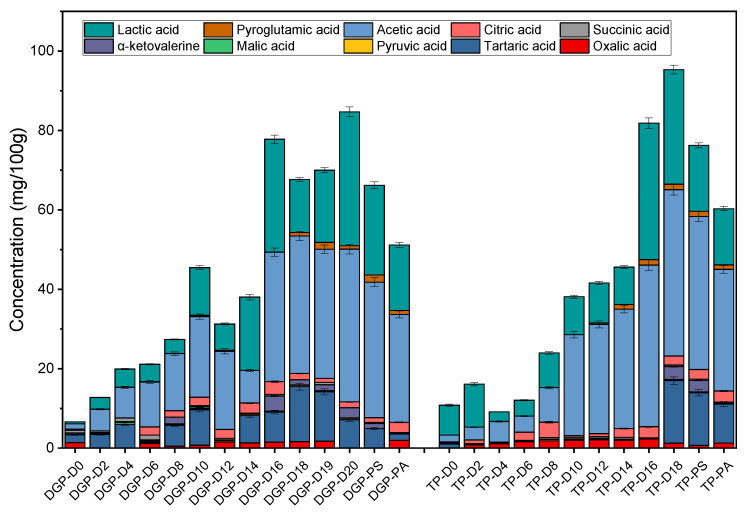
Variations in organic acid during acetic acid fermentation with the traditional process and dry gelatinization process (PS: post-sealing, PA: post-aging).

**Figure 6 foods-13-01071-f006:**
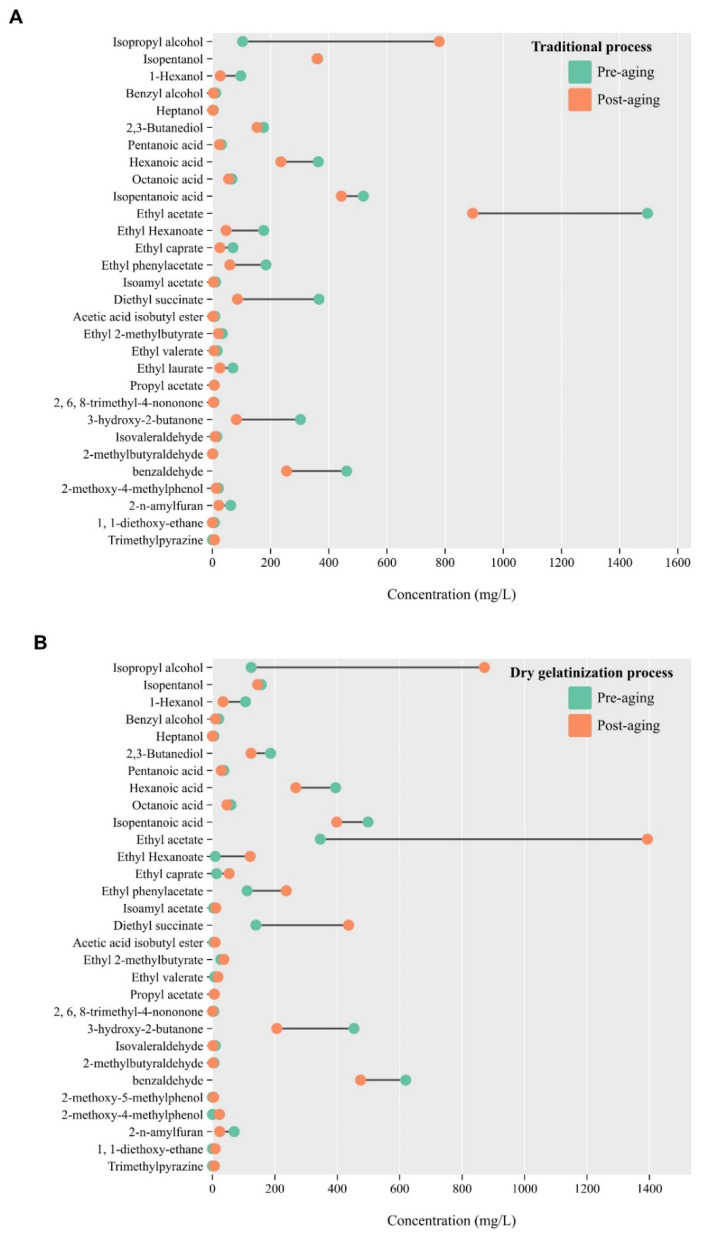
Volatile material content change in the traditional process (**A**) and the dry gelatinization process (**B**).

**Table 1 foods-13-01071-t001:** Parameters of the alcohol fermentation stage.

	Dry Gelatinization Process	Traditional Process
Fermentation period (d)	5	6
Alcohol concentration (% *v*/*v*)	16.2 ± 0.2	13.9 ± 0.5

**Table 2 foods-13-01071-t002:** Flavor analysis of free amino acids in vinegar produced by different processes ^a^.

Free Amino Acids	Threshold Value	DoT Value	
		Dry Gelatinization Process	Traditional Process
Ser	150	1.36	1.09
Thr	260	0.00	15.38
Orn	125	3.76	6.72
Asn	100	12.17	0.00
Glu	30	61.99	18.15
Val	40	18.05	23.14
Leu	190	0.00	0.97
Tyr	NF ^b^	-	-
Phe	90	2.39	3.23

^a^ When DoT is less than 1, it is considered that the substance does not contribute to taste, and when DoT is greater than 1, it is considered that the substance contributes to taste. ^b^ NF, not found.

## Data Availability

The original contributions presented in the study are included in the article, further inquiries can be directed to the corresponding author.

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
