# Peer review of "Investigation of Zhenjiang Aromatic Vinegar Production Using a Novel Dry Gelatinization Process"

_foods, 2024, doi:10.3390/foods13071071_

Round 1

Reviewer 1 Report

Comments and Suggestions for Authors

Letter to Authors 

Dear Authors, 

I have thoroughly reviewed your manuscript entitled "Investigation of Zhenjiang aromatic vinegar production using a novel dry gelatinization process" by Wang et al., submitted to the Foods Journal. Overall, the methodology developed to enhance the flavor profile of Zhenjiang aromatic vinegar is intriguing. Nonetheless, there are several areas that require substantial revisions to meet the standards of Foods Journal. Please see below for the major issues identified within your study:   

         Lines 84-85: Please provide a detailed explanation of dry gelatinization process (DGP) (e.g., technical specifications, etc.) and ensure that the novelty of your study is clearly articulated.  

         Lines 92-96: Please detail more the Methods section (e.g., rationale behind choosing the fermentation period, control measures, replication of experiments, etc.)

         Lines 102-104: The authors fail to state how the fermentation process was controlled in terms of temperature, humidity, air (O2/CO2) levels.

         Line 93: Please detail all the properties and characteristics of the glutinous rice used during this investigation, as these could significantly affect the fermentation outcomes.

         Lines 118-130: Please provide an in-depth investigation regarding the vinegar flavor profiles including the content of each compound produced and its contribution to the overall flavor.

         Throughout the study: The authors fail to investigate the overall impact of dry gelatinization process on the microbiol ecology, which has a significant effect on the vinegar manufacturing.  

         Lines 207-209: The authors did not mention how the yield was determinized (i.e., from the raw materials to the finished products).

         Results Section: This section contains some inconsistencies, ranging from data malpresentation to inadequate comparison of all key fermentation metrics.

         Discussion Section: The economic implication of utilizing DGP, in terms of costs compared to conventional methods, are not discussed.

         It appears the authors have neglected to conduct a sensory evaluation test, which is very crucial for validating claims regarding flavor profile enhancements.

         Comparative storage stability tests on the obtained vinegar samples versus those traditionally manufactured were not performed.  

Comments on the Quality of English Language

         The paper requires an extensive editing for English language and grammar.   

Author Response

  • Lines 84-85: Please provide a detailed explanation of dry gelatinization process (DGP) (e.g., technical specifications, etc.) and ensure that the novelty of your study is clearly articulated. 

Response: A schematic diagram of the equipment for dry gelatinization process is added and the technical specifications are explained. The novelty of our study is reorganized. (Figure 1, Lines 87-115)

  • Lines 92-96: Please detail more the Methods section (e.g., rationale behind choosing the fermentation period, control measures, replication of experiments, etc.)

Response: Materials and Methods section has been detailed more. (Lines 121-191)

  • Lines 102-104: The authors fail to state how the fermentation process was controlled in terms of temperature, humidity, air (O2/CO2) levels.

Response: The control of fermentation process has been stated. (Lines 147-149)

  • Line 93: Please detail all the properties and characteristics of the glutinous rice used during this investigation, as these could significantly affect the fermentation outcomes.

Response: The properties and characteristics of the glutinous rice used during this investigation have been added. (Lines 129-130).

  • Lines 118-130: Please provide an in-depth investigation regarding the vinegar flavor profiles including the content of each compound produced and its contribution to the overall flavor.

Response: By revising the manuscript, we further elucidated the contributions of amino acids, organic acids, and volatile compounds to the flavor profile of vinegar. we also conduct a comparative analysis of compounds that exhibit significant variations between the dry gelatinization process and traditional process. This comparison highlights their respective impacts on the flavor of vinegar, thereby delineating the distinctions in flavor profiles between vinegars produced via the dry gelatinization process and those made through traditional process, thus evaluating the feasibility of the dry gelatinization process.

  • Throughout the study: The authors fail to investigate the overall impact of dry gelatinization process on the microbiol ecology, which has a significant effect on the vinegar manufacturing.

Response: Thank you for your advice. The overall impact of dry gelatinization process on the microbiol ecology will be investigated in future research.

  • Lines 207-209: The authors did not mention how the yield was determinized (i.e., from the raw materials to the finished products).

Response: The usage of “yield” is inappropriate, so it has been revised to production. Given that the quality of the raw materials utilized in both processes remain identical, and taking into account the negligible differences in the acidity levels of the vinegar produced post-leaching, it is plausible to assess the yield of the two processes based on the quantity of vinegar generated following the leaching process. (Lines 276-279)

  • Results Section: This section contains some inconsistencies, ranging from data malpresentation to inadequate comparison of all key fermentation metrics.

Response: The results section has been revised to resolve the inconsistencies. (Results and Discussion section)

  • Discussion Section: The economic implication of utilizing DGP, in terms of costs compared to conventional methods, are not discussed.

Response: The economic implication of utilizing DGP, in terms of costs compared to conventional methods, has been added. (Lines 540-546)

  • It appears the authors have neglected to conduct a sensory evaluation test, which is very crucial for validating claims regarding flavor profile enhancements.

Response: Thank you for your advice. Free amino acids, organic acids, and volatile organic compounds significantly contribute to the complex flavor profile of aromatic vinegars. Consequently, a comparative analysis of vinegars produced by both processes was undertaken, centering on these components as pivotal parameters. Sensory evaluation test will be further conducted in future study.

  • Comparative storage stability tests on the obtained vinegar samples versus those traditionally manufactured were not performed.

Response: Thank you for your advice. The current study preliminarily evaluated the dry gelatinization process from the perspectives of fermentation process parameters and the flavor profile of the vinegar product, and comparative storage stability tests on the obtained vinegar samples versus those traditionally manufactured will be performed in subsequent research work.

Reviewer 2 Report

Comments and Suggestions for Authors

The manuscript describes original research comparing 2 different methods of vinegar production (traditional method and a novel dry gelatinization process) with the aim of addressing some of the challenges faced by the traditional methods such as high-water usage, wastewater generation, raw material losses, and limitations in mechanization and workshop conditions. Though the manuscript reports of differences in the vinegar production processes in terms of efficiency and end product characteristics, the challenges have not been addressed as purported.

Generally, the abstract is clear, and introduction provide background information in line with the study reported.

There is, however, the need for authors to provide some details for the materials and methods section. It would be useful for authors to provide more detail of some of the methods applied even though standards have been referred to. There is also the need to provided specific quantities/volume of samples and other materials used. e.g., Lines 102-108: what is the ratio of water to rice? What amount of marinade is collected on the sampling days? What amount/quantity of samples was applied in the centrifugation process for the analysis of organics acids, free amino acids etc. Similar requirements in other parts.

Most part of the results is just presentation of the results without critical analysis and comparison with literature. To enhance the strength and impart of this manuscript, which will be an excellent read, authors will have to reconsider and enhance the discussion of the results with evidence and support from literature.

Authors have reported results on taste profile analysis as taste threshold values, however, there is no information in the material and methods on how these was conducted and how the taste threshold values were calculated.

There were some parts of the results reported that were not clearly evidence in the results presented and authors will need to check and ensure this is clear.

Lines 149-158: It is not clear how authors arrived at the values of alcohol conversion efficiency and what is the evidence of authors for these claims? What is the evidence of the proliferation of yest cells if authors did not measure this? Similarly, authors to check the following lines to provide evidence of how they arrived at these values: Lines 175-176, 207-210, 235-237.

Line 181: Please, specify the predefined ratio.

Lines 241-245: Proline is not presented in the results, so where is the evidence? If proline is responsible for sweetness taste, one will expect the traditional product to have a sweeter flavour profile than the dry gelatinisation product, which is opposite to what authors have reported.

Lines 306-309: will be useful for authors to provide evidence by way of a reference.

I am not sure why authors decided to use this type of diagram in figures 3 and 4 to present the results. This does not provide a clear view of the results and will suggest author to use a different format.

Comments on the Quality of English Language

Quality of English language is acceptable.

Author Response

The manuscript describes original research comparing 2 different methods of vinegar production (traditional method and a novel dry gelatinization process) with the aim of addressing some of the challenges faced by the traditional methods such as high-water usage, wastewater generation, raw material losses, and limitations in mechanization and workshop conditions. Though the manuscript reports of differences in the vinegar production processes in terms of efficiency and end product characteristics, the challenges have not been addressed as purported.

Generally, the abstract is clear, and introduction provide background information in line with the study reported.

There is, however, the need for authors to provide some details for the materials and methods section. It would be useful for authors to provide more detail of some of the methods applied even though standards have been referred to. There is also the need to provided specific quantities/volume of samples and other materials used. e.g., Lines 102-108: what is the ratio of water to rice? What amount of marinade is collected on the sampling days? What amount/quantity of samples was applied in the centrifugation process for the analysis of organics acids, free amino acids etc. Similar requirements in other parts.

Response: Materials and Methods section has been detailed more. (Lines 120-191).

Most part of the results is just presentation of the results without critical analysis and comparison with literature. To enhance the strength and impart of this manuscript, which will be an excellent read, authors will have to reconsider and enhance the discussion of the results with evidence and support from literature.

Response: The discussion of the results with evidence and support from literature has been enhanced. (Results and Discussion section).

Authors have reported results on taste profile analysis as taste threshold values, however, there is no information in the material and methods on how these was conducted and how the taste threshold values were calculated.

Response: Results on taste profile analysis for amino acid has been revised and corresponding method has been added. (Lines 175-181, 346-362, Table 2)

There were some parts of the results reported that were not clearly evidence in the results presented and authors will need to check and ensure this is clear.

Lines 149-158: It is not clear how authors arrived at the values of alcohol conversion efficiency and what is the evidence of authors for these claims? What is the evidence of the proliferation of yest cells if authors did not measure this? Similarly, authors to check the following lines to provide evidence of how they arrived at these values: Lines 175-176, 207-210, 235-237.

Response:

Lines 149-158: As detailed in the Materials and Methods section, the quality of glutinous rice and the initial moisture content of the alcohol fermentation medium for both processes were identical. The alcohol concentrations of the wine mash produced via the dry gelatinization process and the traditional method registered at 16.2% and 13.9% (v/v), respectively. Notably, the fermentation duration for the dry gelatinization method spanned 5 days, one day shorter than its traditional counterpart. By applying the prescribed formula for production efficiency, the dry gelatinization process was found to surpass the traditional process by 39.1% in terms of production efficiency. However, the challenge in tracking yeast growth during the fermentation phase complicates direct observations. Consequently, the observation that the dry gelatinization method yielded a greater volume of alcohol under identical raw material quality and a reduced fermentation timeline infers a potential acceleration in yeast growth and an enhanced rate of alcohol fermentation within this process. (Lines 205-211)

Lines 175-176: The fermentation cycles for vinegar produced via traditional process and dry gelatinization process spanned 18 and 20 days, respectively. Following the fermentation process, marinade samples were obtained for analysis. Total acidity levels were measured employing the procedure outlined in the Materials and Methods section, yielding concentrations of 7.178% (w/v) and 7.092% (w/v) for the traditional and dry gelatinization process, respectively. Moreover, following the sealing of the fermentation vessel, the rate of acidity in the dry gelatinization process consistently exceeded that of the traditional process. This phenomenon can be linked to the post-alcohol fermentation step, wherein water was added to the mash to adjust the alcohol concentration to approximately 9% (v/v). Subsequently, a mixture of wheat bran and rice husk was added to the mash in a predefined ratio according to vinegar production procedures. The dry gelatinization process, having a 14% higher alcohol content than the traditional process, resulted in a significantly larger initial mash volume post-dilution. Consequently, the initial moisture content in the vinegar pei from the dry gelatinization process was considerably higher compared to the traditional process. This disparity influenced the oxygen availability to acetic acid bacteria during acetic acid fermentation, thereby affecting acid production. After vessel sealing, anaerobic lactic acid bacteria, unaffected by oxygen levels, produced increased amounts of lactic acid. Thus, the total acid content in the dry gelatinization process surpassed that of the traditional process post-sealing.

    In the Materials and Methods section, the usage of “total acid content” is inappropriate, so it has been revised to “total acidity”. (Lines 160-161)

Lines 207-210: The usage of “yield” is inappropriate, so it has been revised to production. Given that the quality of the raw materials utilized in both processes remains identical, and taking into account the negligible differences in the acidity levels of the vinegar produced post-leaching, it is plausible to assess the yield of the two processes based on the quantity of vinegar generated following the leaching process. (Lines 276-279)

Lines 235-237: The total amino acid concentrations after acetic acid fermentation were the cumulative concentration of amino acids detected in the marinade samples. This detail has been duly added to the Materials and Methods section to ensure clarity. (Lines 181-183)

Line 181: Please, specify the predefined ratio.

Response: The predefined ratio has been specified. (Lines 245-246)

Lines 241-245: Proline is not presented in the results, so where is the evidence? If proline is responsible for sweetness taste, one will expect the traditional product to have a sweeter flavour profile than the dry gelatinisation product, which is opposite to what authors have reported.

Response: The two sentences were written incorrectly and they have been revised. (Lines 319-323)

Lines 306-309: will be useful for authors to provide evidence by way of a reference.

Response: A reference has been provided. (Lines 401-403)

I am not sure why authors decided to use this type of diagram in figures 3 and 4 to present the results. This does not provide a clear view of the results and will suggest author to use a different format.

Response: Figures 3 and 4 have been revised. (Figures 4 and 5)

Reviewer 3 Report

Comments and Suggestions for Authors

The topic approached could have a significant practical applicability. It also has a durability dimension through the contribution brought by decreasing the water needed and the resulting wastewater. Another aspect that raises attention is the increase in alcohol conversion yield. The results obtained are promising and might stimulate future developments, as the authors highlighted.

To improve the scientific level of the manuscript, the researchers could consider including some bibliographic references to support some of their statements:

·         amino acids….”They play a crucial role in cellular metabolism regulation and function as bioactive compounds, bolstering immunity and facilitating brain development.”-ln. 223-224

·         When ingested, vinegar-derived amino acids display biological and metabolic characteristics akin to free amino acids, forming complex peptides such as immunoglobulins, carrier proteins, and neurotransmitters.-ln. 225-226

·         Notably, certain amino acids in vinegar, like histidine, methionine, cysteine, tryptophan, and tyrosine, exhibit potent antioxidant properties.-ln 227

·         Conversely, lactic, tartaric, malic, and succinic acids serve as buffering agents, moderating the intensity of acetic acid and thus contributing to a more balanced flavor.-ln 301-302

·         Vinegar's organic acids are widely recognized for their health benefits, including antibacterial properties, reducing fat accumulation and hyperlipidemia, improving insulin resistance and metabolic disorders, lowering high blood pressure, and mitigating fatigue.-ln. 306-308

·         The concentration of these organic acids in vinegar varies significantly, influenced by the vinegar type and the specifics of the fermentation process.-ln. 309-310

  The paper could be accepted for publication after minor changes. It has to be revised by the authors and resubmitted with suggested modifications specified in the reviewer’s comments.

Comments on the Quality of English Language

 Minor editing of English language required.

Author Response

The topic approached could have a significant practical applicability. It also has a durability dimension through the contribution brought by decreasing the water needed and the resulting wastewater. Another aspect that raises attention is the increase in alcohol conversion yield. The results obtained are promising and might stimulate future developments, as the authors highlighted.

To improve the scientific level of the manuscript, the researchers could consider including some bibliographic references to support some of their statements:

  • amino acids….”They play a crucial role in cellular metabolism regulation and function as bioactive compounds, bolstering immunity and facilitating brain development.”-ln. 223-224
  • When ingested, vinegar-derived amino acids display biological and metabolic characteristics akin to free amino acids, forming complex peptides such as immunoglobulins, carrier proteins, and neurotransmitters.-ln. 225-226
  • Notably, certain amino acids in vinegar, like histidine, methionine, cysteine, tryptophan, and tyrosine, exhibit potent antioxidant properties.-ln 227
  • Conversely, lactic, tartaric, malic, and succinic acids serve as buffering agents, moderating the intensity of acetic acid and thus contributing to a more balanced flavor.-ln 301-302
  • Vinegar's organic acids are widely recognized for their health benefits, including antibacterial properties, reducing fat accumulation and hyperlipidemia, improving insulin resistance and metabolic disorders, lowering high blood pressure, and mitigating fatigue.-ln. 306-308
  • The concentration of these organic acids in vinegar varies significantly, influenced by the vinegar type and the specifics of the fermentation process.-ln. 309-310

Response 3: Bibliographic references have been added to support above statements.

  The paper could be accepted for publication after minor changes. It has to be revised by the authors and resubmitted with suggested modifications specified in the reviewer’s comments.

Round 2

Reviewer 1 Report

Comments and Suggestions for Authors

Dear Authors,

I am very satisfied with the revision. The paper can be now published in its current form.

Congratulations!